# OUT-OF-CLASS NOVELTY GENERATION: AN EXPERIMENTAL FOUNDATION

**Mehdi Cherti & Balázs Kégl**
LAL/LRI
CNRS/Université Paris-Saclay
{mehdi.cherti, balazs.kegl}@gmail.com

**Akın Kazakçı**
MINES ParisTech,
PSL Research University, CGS-I3 UMR 9217
akin.kazakci@mines-paristech.fr

## ABSTRACT

Recent advances in machine learning have brought the field closer to computational creativity research. From a creativity research point of view, this offers the potential to study creativity in relationship with knowledge acquisition. From a machine learning perspective, however, several aspects of creativity need to be better defined to allow the machine learning community to develop and test hypotheses in a systematic way. We propose an actionable definition of creativity as the generation of out-of-distribution novelty. We assess several metrics designed for evaluating the quality of generative models on this new task. We also propose a new experimental setup. Inspired by the usual held-out validation, we hold out entire classes for evaluating the generative potential of models. The goal of the novelty generator is then to use training classes to build a model that can generate objects from future (hold-out) classes, unknown at training time - and thus, are novel with respect to the knowledge the model incorporates. Through extensive experiments on various types of generative models, we are able to find architectures and hyperparameter combinations which lead to out-of-distribution novelty.

## 1 INTRODUCTION

Recent advances in machine learning have renewed interest in artificial creativity. Studies such as deep dream (Mordvintsev et al., 2015) and style transfer (Gatys et al., 2015) have aroused both general public interest and have given strong impetus to use deep learning models in computational creativity research (ICC, 2016). Although creativity has been a topic of interest on and off throughout the years in machine learning (Schmidhuber, 2009), it has been slowly becoming a legitimate sub-domain with the appearance of dedicated research groups such as Google's Magenta and research work on the topic (Nguyen et al., 2015; Lake et al., 2015).

There is a large body of work studying creativity by computational methods. A large variety of techniques, from rule-based systems to evolutionary computation has been used for a myriad of research questions. Compared to these methods, machine learning methods provide an important advantage: they enable the study of creativity in relation with knowledge (i.e., knowledge-driven creativity; Kazakçı et al. (2016)). Nevertheless, to better highlight the points of interest in computational creativity research for the machine learning community and to allow machine learning researchers to provide systematic and rigorous answers to computational creativity problems, it is important to precisely answer three questions:

1. What is meant by the generation of novelty?
2. How can novelty be generated?
3. How can a model generating novelty be evaluated?

Within the scope of machine learning, it would be tempting to seek answers to these questions in the sub-field on generative modeling. Mainstream generative modeling assumes that there is a phenomena generating the observed data and strive to build a model of that phenomena, which would, for instance, allow generating further observations. Traditional generative modeling considers only *in-distribution* generation where the goal is to generate objects from the category or categories of

already observed objects. In terms of novelty generation, this can be considered as generating look-a-likes of known *types* of objects. Although there is considerable value in in-distribution generation (e.g., for super-resolution (Freeman et al., 2002; Dong et al., 2014; Ledig et al., 2016) or in-painting (Xie et al., 2012; Cho, 2013; Yeh et al., 2016)), this perspective is limited from a strict point of view of creativity: it is *unlikely* to come up with a *flying ship* by generating samples from a distribution of *ships* and *flying objects*.

Researchers in creativity research (Runco & Jaeger, 2012) have argued that the crux of creative process is the ability to build new categories based on already known categories. However, creativity is beyond a simple combination exploration: it is about generating previously unknown but meaningful (or valuable) new types of objects using previously acquired knowledge (Hatchuel & Weil, 2009; Kazakçı, 2014). Under this perspective, novelty generation aims at exhibiting an example from a new type. This objective, which we shall call *out-of-distribution generation*, is beyond what can be formalized within the framework of traditional learning theory, even though learning existing types is a crucial part of the process.

From a machine learning point of view, generating an object from an unknown type is not a well-defined problem, and research in generative modeling usually aims at *eliminating* this possibility altogether, as this is seen as a source of instability (Goodfellow et al., 2014; Salimans et al., 2016) leading to spurious samples (Bengio et al., 2013). In a way, sampling procedures are designed to kill any possibility of sampling out of the distribution, which is a problem for studying the generation of novelty by machine learning methods.

Arguably, the most important problem is the evaluation of what constitutes a good model for generating out-of-distribution. On the one hand, we are seeking to generate *meaningful* novelty, not trivial noise. On the other hand, we aim at generating *unknown* objects, so traditional metrics based on the concept of likelihood are of no use since novelty in the out-of-distribution sense is unlikely by definition. This lack of metrics hinders answering the first two questions. Without a clear-cut evaluation process, the utility of extending the definition of novelty generation to out-of-sample seems pointless.

This paper argues that for a wider adoption of novelty generation as a topic for scientific study within machine learning, a new engineering principle is needed, which would enable such evaluation, and consequently, rigorous experimental research. In the traditional supervised context, the main engineering design principle is the minimization of the error on a hold-out test set. The paper proposes a simple setup where the generative potential of models can be evaluated by *holding out entire classes*, simulating thus unknown but meaningful novelty. The goal of the novelty generator is then to use training classes to build a model that can generate objects from future (hold-out) classes, unknown at training time.

The main contributions of this paper:

- We design an experimental framework based on hold-out classes to develop and to analyze out-of-distribution generators.

- We review and analyze the most common evaluation techniques from the point of view of measuring out-of-distribution novelty. We argue that likelihood-based techniques inherently limit exploration and novelty generation. We carefully select a couple of measures and demonstrate their applicability for out-of-distribution novelty detection in experiments.

- We run a large-scale experimentation to study the ability of novelty generation of a wide set of different autoencoders and GANs. The goal here is to re-evaluate existing architectures under this new goal in order to open up exploration. Since out-of-distribution novelty generation is arguably a wider (and softer) objective than likelihood-driven sampling from a fixed distribution, existing generative algorithms, designed for this latter goal, constitute a small subset of the algorithms able to generate novelty. The goal is to motivate the reopening some of the closed design questions.

The paper is organized as follows. We review some of the seminal work at the intersection of machine learning and out-of-distribution generation in Section 2. We discuss the conceptual framework of out-of-distribution generation and its relationship with likelihood-based generative models in Section 3. We outline the families of evaluation metrics, focusing on those we use in the paper in Section 4. In Section 4.3 we describe the gist of our experimental setup needed to understand the

metrics described in Section 4.4, designed specifically for the out-of-distribution setup. We describe the details of the experimental setup and analyze our results in Section 5. Finally, we conclude in Section 6.

The paper can be read either in order of the sections, first the motivation and conceptual underpinning of the framework, then the technical contribution, or the other way around, by jumping the Section 4, then coming back to Sections 2 and 3.

## 2 MACHINE LEARNING AND NOVELTY GENERATION: THE INNOVATION ENGINE, "ZERO-SHOT" LEARNING, AND DISCOVERING NEW TYPES

There are three important papers that consider novelty generation in a machine learning context. Nguyen et al. (2015) propose an innovation engine (Figure 1(a)). They generate images using a neural net that composes synthetic features. The generator is fed back with an entropy-based score (similar to objectness; Section 4.2) coming from an Imagenet classifier, and the feedback is used in an evolutionary optimization loop to drive the generation. An important contribution of the paper is to demonstrate the importance of the objectness score. They show that interesting objects are not generated when asking the machine to generate from a single given class. The generation paths often go through objects from different classes, "stepping stones" which are seemingly unrelated to the final object. The main conceptual difference between our approaches is that Nguyen et al. (2015) do not ground their generative model in learned knowledge: their generation process is not learned model, rather a stochastic combinatorial engine. On the one hand, this makes the generation (evolutionary optimization) rather slow, and on the other, the resulting objects reflect the style of the (preset) synthetic features rather than features extracted from existing objects.

The main goal of Lake et al. (2015) and Rezende et al. (2016) is *one-shot learning and generation*: learn to classify objects given a small number (often one) of examples coming from a given category, and learn to generate new objects given a single example (Figure 1(b)). One-shot generation is definitely an intermediate step towards out-of-distribution generation. The extremely low number of examples conceptually limits likelihood-based learning/fitting/generation. Lake et al. (2015) circumvents this problem by learning strong Bayesian top-down models (programs) that capture the structural properties of known objects which are generalizable across classes. They also consider unconstrained ("zero-shot") generation as an extension of their approach, and show that the model can generate new symbols from scratch. They make no attempt to conceptualize the goal of unconstrained generation outside the top-down Bayesian framework, or to design evaluation metrics to assess the quality of these objects, but their intriguing results are one of the strongest motivations of our paper.

Kazakçı et al. (2016) show that symbols of new types can be generated by carefully tuned autoencoders, learned entirely bottom-up, without imposing a top-down Bayesian architecture (Figure 1(c)). They also make a first step of defining the conceptual framework of novelty generation by arguing the goal of generating objects from new *types*, unknown at the time of training. They design a technique for finding these new types semi-automatically (combining clustering and human labeling). They argue the importance of defining the *value* of these new types (and of out-of-distribution generation in general), but they make no attempt to design evaluation metrics, thus limiting the exploration and the development of out-of-distribution generative architectures.

## 3 PROBABILISTIC VS. CONSTRUCTIVE GENERATIVE MODELS

The generative process is commonly framed in a probabilistic setup: it is assumed that an underlying unknown likelihood *model* $\mathcal{P}(\cdot)$ should first be learned on an i.i.d. *training* sample $\mathcal{D} = \{\mathbf{x}_1, \ldots, \mathbf{x}_n\}$, assumed to be generated from $\mathcal{P}(\cdot)$, and then a *sampler* $\mathcal{S}$ should sample from the learned $\widehat{\mathcal{P}}(\cdot)$. The first step, estimating $\mathcal{P}(\cdot)$ using $\mathcal{D}$, is a classical function learning problem that can be studied through the usual concepts of overfitting and regularization, and algorithms can be designed using the classical train/test principle. The second step, designing $\mathcal{S}$ for sampling from $\widehat{\mathcal{P}}(\cdot)$ is also a classical domain of random sampling with a conceptual framework and a plethora of methods.

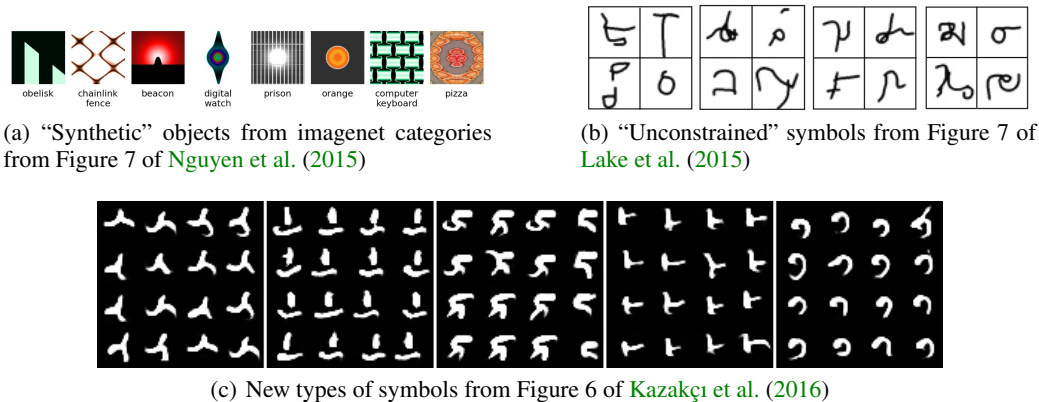

(a) "Synthetic" objects from imagenet categories from Figure 7 of Nguyen et al. (2015)

(b) "Unconstrained" symbols from Figure 7 of Lake et al. (2015)

(c) New types of symbols from Figure 6 of Kazakçı et al. (2016)

Figure 1: Examples of generating new objects or types.

Technically both steps are notoriously hard for the high-dimensional distributions and the complex dependencies we encounter in interesting domains. Hence, most of the recent and successful methods get rid of the two-step procedure at the level of algorithmic design, and short-cut the procedure from the probabilistic $\mathcal{D} \rightarrow \mathcal{P} \rightarrow \mathcal{S}$ to the constructive $\mathcal{D} \rightarrow \mathcal{A}$, where $\mathcal{A}(\mathcal{D})$ is a *generator*, tasked to produce sample objects *similar* to elements of $\mathcal{D}$ but *not identical* to them. $\mathcal{A}$ is fundamentally different from $(\mathcal{P}, \mathcal{S})$ in that there is no explicit fitting of a *function*, we use $\mathcal{D}$ to directly *design an algorithm* or a *program*.

When the probabilistic setup is still kept for *analysis*, we face a fundamental problem: if we assume that we are given the true likelihood function $\mathcal{P}(\cdot)$, the likelihood of the training sample $\frac{1}{n}\sum_{i=1}^{n}\log\mathcal{P}(\mathbf{x}_i)$ is a random variable drawn independently from the distribution of log-likelihoods of i.i.d. samples of size $n$, so the trivial generator $\mathcal{A}$ which *resamples* $\mathcal{D}$ will have the same expected log-likelihood as an optimal i.i.d. sampler. The resampling "bug" is often referred to as "overfitting". While it makes perfect sense to talk about overfitting in the $\mathcal{D} \rightarrow \mathcal{P} \rightarrow \mathcal{S}$ paradigm (when $\mathcal{P}$ is *fitted* on $\mathcal{D}$), it is somewhat conceptually misleading when there is no fitting step, we propose to call it "memorizing". When a generator $\mathcal{A}$ is trained on $\mathcal{D}$ without going through the fitting step $\mathcal{D} \rightarrow \mathcal{P}$, the classical tools for avoiding memorizing (regularization, the train/test framework) may be either conceptually inadequate or they may not lead to an executable engineering design principle.

The conceptual problem of analyzing constructive algorithms in the probabilistic paradigm is not unrelated to our argument of Section 1 that the probabilistic generative framework is too restrictive for studying novelty generation and for designing out-of-distribution generative models. In our view, this flaw is not a minor nuisance which can be fixed by augmenting the likelihood to avoid resampling, rather an inherent property which cannot (or rather, should not) be fixed. The probabilistic framework is designed for generating objects from the distribution of known objects, and this is in an axiomatic contradiction with generating out-of-distribution novelty, objects that are *unknown* at the moment of assembling a training sample. Resampling (generating *exact* copies) is only the most glaring demonstration of a deeper problem which is also present in a more subtle way when attempting to generate new *types* of objects.

We are not arguing that the probabilistic generative framework should be banished, it has a very important role in numerous use cases. Our argument is that it is not adequate for modeling out-of-distribution novelty generation. What follows from this on the *algorithmic* level is not revolutionary: the design of most successful generative algorithms already moved beyond the probabilistic framework. On the other hand, moving beyond the probabilistic generative framework at a *conceptual* level is a paradigm change which will require groundwork for laying the foundations, including revisiting ideas from a domain larger than machine learning.

At the algorithmic/computational level the machine learning community has already started to move beyond likelihood. The overfitting problem is often solved by implicitly constraining $\mathcal{A}$ not to resample. Another common solution is to design tractable likelihood surrogates that implicitly penalize memorization. These surrogates then can be used at the training phase (to obtain non-resampling

generators explicitly) and/or in the evaluation phase (to eliminate generators that resample). The ingenious idea of using discriminators in GANs (Goodfellow et al., 2014; Salimans et al., 2016) is a concrete example; although the setup *can* be analyzed through the lens of probabilistic sampling, one does not *have to* fall back onto this framework. If we drop the underlying conceptual *probabilistic* framework, the *constructive* GAN idea may be extended beyond generating from the *set* which is indistinguishable from the set of existing objects. In Section 4.4 we will use discriminators to assess the quality of generators whose very goal is to generate novelty: objects that *are* distinguishable from existing objects. The main challenge is to avoid the trivial novelty generator, producing uninteresting noise. This challenge is structurally similar to avoiding the trivial memorizing/resampling generator in in-distribution sampling. The two main elements that contribute to the solution is i) to ground the generator strongly in the structure of existing *knowledge*, without overly fixating it on existing *classes*, and ii) use a discriminator which knows about *out-of-class* novelty to steer architectures towards novelty generation.

# 4 EVALUATION OF GENERATIVE MODELS

In this section we outline the families of evaluation metrics, focusing on those we use in the paper. In Section 4.3 we describe the gist of our experimental setup needed to understand the metrics described in Section 4.4, designed specifically for the out-of-distribution setup.

## 4.1 INDIRECT SUPERVISED METRICS

When generative models are used as part of a pipeline with a supervised goal, the evaluation is based on the evaluation of the full pipeline. Examples include unsupervised pre-training (Hinton et al. (2006); Bengio et al. (2007); the original goal that reinvigorated research in neural nets), semi-supervised learning (Kingma et al., 2014; Rasmus et al., 2015; Maaløe et al., 2016; Salimans et al., 2016), in-painting (Xie et al., 2012; Cho, 2013; Yeh et al., 2016), or super-resolution (Freeman et al., 2002; Dong et al., 2014; Ledig et al., 2016). The design goal becomes straightforward, but the setup is restricted to improving the particular pipeline, and there is no guarantee that those objectives can be transferred between tasks. In our case, the objective of the supervised pipeline may actually suppress novelty. In a certain sense, GANs also fall into this category: the design goal of the generator is to fool a high-quality discriminator, so the generator is asked *not* to generate new objects which can be easily discriminated from known objects. In our experiments, surprisingly, we found that GANs can be still tuned to generate out-of-distribution novelty, probably due to the deficiencies of both the generator and the discriminator. Our goal in this paper can also be understood as designing a pipeline that turns novelty generation into a supervised task: that of generating objects from classes unknown at training time.

### 4.1.1 PARZEN DENSITY ESTIMATOR

Parzen density estimators are regularly used for estimating the log-likelihood of a model (Breuleux et al., 2009). A kernel density estimator is fit to generated points, and the model is scored by log-likelihood of a hold-out test set under the kernel density. The metrics can be easily fooled (Theis et al., 2015), nevertheless, we adopted it in this paper for measuring both the in-distribution and out-of-distributions quality of our generators.

## 4.2 OBJECTNESS

Salimans et al. (2016) proposed a new entropy-based metrics to measure the "objectness"[1] of the generated *set* of objects. As GANs, the metrics uses a trained discriminator, but unlike GANs, it is not trained for separating real objects and generated objects, rather to classify real objects into existing categories. The goal of the generator is create objects which belong confidently to a low number (typically one) of classes. To penalize generators fixating onto single objects or categories, they also require that the *set* of objects has a high entropy (different objects span the space of the categories represented by the discriminator). The metrics is only indirectly related to classical log-likelihood: in a sense we measure how likely the objects are *through the "eye" of a discriminator*.

---

[1] They also call it "inception score" but we found the term objectness better as it is more general than the single model used in their paper.

Formally, objectness is defined as

$$\frac{1}{N}\sum_{i=1}^{n}\sum_{\ell=1}^{K} p_{i,\ell}\log\frac{p_{i,\ell}}{p_\ell},$$

where $K$ is the number of classes,

$$p_{i,\ell} = \mathcal{P}(\ell|\mathbf{x}_i)$$

is the posterior probability of category $\ell$ given the generated object $\mathbf{x}_i$, under the discriminator $\mathcal{P}$ trained on a set with known labels, and

$$p_\ell = \frac{1}{n}\sum_{i=1}^{n} p_{i,\ell},$$

are the class marginals.

Salimans et al. (2016) proposed this metric as one of the "tricks" to stabilize GANs, but, interestingly, a similar measure was also used in the context if evolutionary novelty generation (Nguyen et al., 2015).

### 4.3 Assessing out-of-distribution novelty by out-of-class scoring

As the classical supervised validation setup simulates past (training) and future (test) by randomly partitioning an existing data set, we can simulate existing knowledge and novelty by partitioning existing data sets *holding out entire classes*. The goal of the novelty generator is then to use training classes to build a model that can generate objects from future (hold-out) classes, unknown at training. In our first experiments we tried to leave out single classes of MNIST, but the label noise "leaked" hold-out classes which made the evaluation tricky. To avoid this, we decided to challenge the generator, trained on MNIST, to generate *letters*. We pre-trained various discriminators using different setups, only on digits (MNIST), only on letters (Google fonts), or on a mixture of digits and letters, and used these discriminators to evaluate novelty generators in different ways. For example, we measure *in-class objectness* and *in-class Parzen* using a discriminator trained on MNIST, and *out-of-class objectness* and *out-of-class Parzen* by a discriminator trained on (only) Google fonts.

### 4.4 Out-of-class scores

Naturally, letter discriminators see letters everywhere. Since letters are all they know, they classify everything into one of the letter classes, quite confidently (this "blind spot" phenomenon is exploited by Nguyen et al. (2015) for generating "synthetic" novelty), the letter objectness of an in-distribution digit generator can sometimes be high. For example, a lot of 6s were classified as $b$s. To avoid this "bias", we also trained a discriminator on the union of digits and letters, allowing it to choose digits when it felt that the generated object looked more like a digit. We designed two metrics using this discriminator: *out-of-class count* measures the frequency of confidently classified letters in a generated set, and *out-of-class max* is the mean (over the set) of the probability of the most likely letter. None of these metrics penalize "fixated" generators, outputting the same few letters all the time, so we combine both metrics with the entropy of the letter posterior (conditioned on being a letter).

Formally, let $p_{i,1},\ldots,p_{i,K_{\text{in}}}$ be the in-class posteriors and $p_{i,K_{\text{in}}+1},\ldots,p_{i,K_{\text{in}}+K_{\text{out}}}$ be the out-of-class posteriors, where $K_{\text{in}} = 10$ is the number of in-class classes (digits), and $K_{\text{out}} = 26$ is the number of out-of-class classes (letters). Let

$$\ell_i^* = \arg\max_\ell p_{i,\ell}$$

and

$$\ell_{\text{out}\,i}^* = \arg\max_{K_{\text{in}} < \ell \le K_{\text{in}}+K_{\text{out}}} p_{i,\ell}$$

be the most likely category overall and most likely out-of-class category, respectively. Let

$$\tilde{p}_\ell = \frac{\sum_{i=1}^{n}\mathbb{I}\{\ell = \ell_{\text{out}\,i}^*\}}{\sum_{i=1}^{n}\mathbb{I}\{\ell_{\text{out}\,i}^* > K_{\text{in}}\}}$$

be the normalized empirical frequency of the out-of-class category $\ell$. We measure the diversity of the generated sample by the normalized entropy of the empirical frequencies

$$\text{diversity} = -\frac{1}{\log K_{\text{out}}} \sum_{\ell=K_{\text{in}}}^{K_{\text{in}}+K_{\text{out}}} \tilde{p}_\ell \log \tilde{p}_\ell,$$

and define

$$\text{out-of-class count} = (1-\lambda) \times \frac{1}{n} \sum_{i=1}^{n} \mathbb{I}\left\{\ell_i^* > K_{\text{in}} \wedge p_{i,\ell_i^*} > \theta\right\} + \lambda \times \text{diversity},$$

and

$$\text{out-of-class max} = (1-\lambda) \times \frac{1}{n} \sum_{i=1}^{n} p_{i,\ell_{\text{out}\,i}^*} + \lambda \times \text{diversity}.$$

In our experiments we set the confidence level $\theta = 0.95$ and the mixture coefficient $\lambda = 0.5$.

## 4.5 HUMAN REFEREEING AND THE VISUAL TURING TEST

The ultimate test of l'art pour l'art generative models is whether humans like the generated objects. Visual inspection is often used as an evaluation principle in papers (Denton et al., 2015; Radford et al., 2015; Dosovitskiy et al., 2016), and it is sometimes even made part of the objectified pipeline by using crowdsourcing tools (Denton et al., 2015; Lake et al., 2015; Salimans et al., 2016). First, it definitely makes development (e.g., model selection and hyperparameter tuning) slow. Second, the results depend a lot on what questions are asked and how the responders are primed. For testing generative models, the usual GAN-type question to ask is whether the generated objects are generated by a nature (or a human) or a machine (the visual Turing test). Even those that go the furthest in tasking machines to generate novelty (Lake et al., 2015) ask human judges to differentiate between human and machine. In our view, this question is too restrictive when the goal is out-of-distribution novelty generation. Asking whether an object is "new" is arguably too vague, but inventing adjective categories (such as "surprising" or "interesting" (Schmidhuber, 2009)) that can poll our ability to detect novelty should be on the research agenda. Priming is another important issue: the answer of a human annotator can depend on the information given to her. Nevertheless, a human annotation tool with well-designed priming and questions could accelerate research in novelty generation in the same way labeling tools and standard labeled benchmark sets accelerated supervised learning.

We assessed the visual quality of the set of generated objects using an in-house annotation tool. We took each model which appeared in the top ten by any of the quantitative metrics described in the previous section, and hand-labeled them into one of the following three categories: i) letters, ii) digits, and iii) bad sample (noise or not-a-symbol).

Each panel consisted $26 \times 15$ generated objects, the fifteen most probable symbols of each letter according to the classifier trained on both letters and digits (Figure 2). The goal of this annotation exercise was i) to assess the visual quality of the generated symbols and ii) to assess the quality of the metrics in evaluating novelty.

## 5 EXPERIMENTS

Our scores cannot be directly optimized because they all measure out-of-class performance, and showing out-of-class objects at training would be "cheating". All our (about 1000) models were trained for "classical" objectives: reconstruction error in the case of autoencoders, and adversarial error in the case of GANs. The out-of-class scores were used as a weak feedback for model selection and (quasi random) hyperparameter optimization. The goal is not to be statistically flawless, after all we do not have a statistical model. Rather we set our goal to analyze existing generative architectures from the point of view of novelty generation. Most of the generative models come from a large class of architectures, sometimes purposefully designed for not to "misbehave". When possible, we turned these tricks, designed to avoid generating "spurious" objects, into optional hyperparameters.

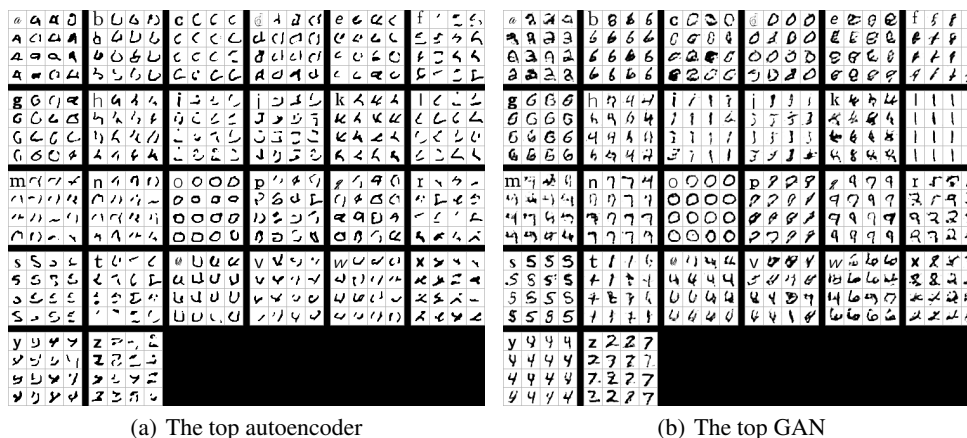

(a) The top autoencoder (b) The top GAN

Figure 2: A couple of the top models according to human assessment. Top left characters of each $4 \times 4$ panel are the labels, letters coming from the training sample. For each letter we display the fifteen most probable symbols according to the classifier trained on both letters and digits.

## 5.1 DETAILED EXPERIMENTAL SETUP

We used two families of deep learning based generative models, autoencoders and GANs. The architectures and the optional features are described in the next sections. All hyperparameters were selected randomly using reasonable priors. All the $\sim$1000 autoencoders were trained on MNIST training data.

### 5.1.1 AUTOENCODER ARCHITECTURES AND GENERATION PROCEDURE

We used three regularization strategies for autoencoders: sparse autoencoders (Makhzani & Frey, 2013; 2015), denoising autoencoders (Bengio et al., 2013) and contractive autoencoders (Rifai et al., 2011).

Sparse autoencoders can either be fully connected or convolutional. For fully connected sparse autoencoders, we use the $k$-sparse formulation from Makhzani & Frey (2013), a simple way of obtaining a sparse representation by sorting hidden units and keeping only the top $k\%$, zeroing out the others, and then backpropagating only through non-zero hidden units.

For convolutional sparse architectures, we use the "winner take all" (WTA) formulation from Makhzani & Frey (2015) which obtains *spatial sparsity* in convolutional feature maps by keeping only the maximum activation of each feature map, zeroing out the others. We optionally combine it with *channel sparsity* which, for each position in the feature maps, keeps only the maximum activation across the channels and zero out the others.

For contractive autoencoders, we use the fully connected version with a single hidden layer from Rifai et al. (2011).

We also explore mixtures between the different autoencoder variants in the hyperparameter search. For each model we choose to enable or disable independently the denoising training procedure, the contractive criterion (parametrized by the contractive coefficient, see (Rifai et al., 2011)) and the sparsity rate $k$ (only for fully connected architectures). Table 1 shows the hyperparameters and their priors.

The generation procedure we use for autoencoders is based on Bengio et al. (2013), who proposed a probabilistic interpretation of denoising autoencoders and a way to sample from them using a Markov chain. To have a convergent procedure and to obtain fixed points, we chose to use a deterministic generation procedure instead of a Markov chain (Bahdanau & Jaeger, 2014). As in Bahdanau & Jaeger (2014), we found that the procedure converged quickly.

In initial experiments we found that 100 iterations were sufficient for the majority of models to have convergence so we chose to fix the maximum number of iterations to 100. We also chose to extend

the procedure of Bahdanau & Jaeger (2014) by binarizing (using a threshold) the images after each reconstruction step, as we found that it improved the speed of the convergence and could lead to final samples with an exact zero reconstruction error.

For stochastic gradient optimization of the autoencoder models, we used adadelta (Zeiler, 2012) with a learning rate of 0.1 and a batch size of 128. We used rectified linear units as an activation function for hidden layers in all models. We use the sigmoid activation function for output layers.

Table 1: Autoencoder hyperparameter priors.

| Name | Prior | Type |
|---|---|---|
| nb layers | 1, 2, 3, 4, 5 | choice |
| nb fully connected hidden units | 100,200,300,...1000 | choice |
| nb conv layers | 1, 2, 3, 4, 5 | choice |
| nb conv filters | 8, 16, 32, 64, 128, 256, 512 | choice |
| conv layers filter size | 3 or 5 | choice |
| noise corruption | [0, 0.5] | uniform |
| k sparsity rate | [0, 1] | uniform |
| contraction coefficient | [0, 100] | uniform |

### 5.1.2 GENERATIVE ADVERSARIAL NETWORKS (GANS)

For GANs, we built upon Radford et al. (2015) and used their architecture as a basis for hyperparameter search. We modified the code proposed here to sample new combinations of hyperparameters. Table 2 shows the hyperparameters and their priors.

| Name | Prior | Type |
|---|---|---|
| nb discr. updates | 1, 2, 3 | choice |
| l2 coeficient | $[10^{-6}, 10^{-1}]$ | logspace |
| gen. input dim. | 10, 20, 50, 70, 100, 150, 200, 300 | choice |
| nb fully connected gen. units | 8, 16, 32, 64, 128, 256, 1024, 2048 | choice |
| nb fully connected discr. units | 8, 16, 32, 64, 128, 256, 1024, 2048 | choice |
| nb filters gen. | 8, 16, 32, 64, 128, 256, 512 | choice |
| nb filters discr. | 8, 16, 32, 64, 128, 256, 512 | choice |
| nb iterations | 50, 100, 150, 200, 250, 300 | choice |
| learning rate | $[10^{-6}, 10^{-1}]$ on logspace, or 0.0002 | logspace |
| weight initialization | Normal(0, std) where std is from $[10^{-3}, 10^{-1}]$ | logspace |

Table 2: GAN hyperparameter priors.

### 5.2 ANALYSIS

First, we found that tuning (selecting) generative models for in-distribution generation will make them "memorize" the classes they are trained to sample from. This is of course not surprising, but it is important to note because it means that out-of-class generation is non-trivial, and the vast majority of architectures designed and tuned in the literature are not generating out-of-class novelty naturally. Second, we did succeed to find architectures and hyperparameter combinations which lead to out-of-class novelty. Most of the generated objects, of course, were neither digits nor letters (Figure 3), which is why we needed the "supervising" discriminators to find letter-like objects among them. The point is not that *all* new symbols are letters, that would arguably be an impossible task, but to demonstrate that by opening up the range of generated objects, we do not generate noise, rather objects that *can be* forming new categories.

The quantitative goal of this study was to assess the quality of the defined *metrics* in evaluating out-of-distribution generators. We proceeded in the following way. We selected the top ten autoencoders and GANs according to the five metrics of out-of-class (letters) count, out-of-class max, out-of-class objectness, out-of-class Parzen, and in-class Parzen. We then annotated these models into one of the three categories of "letter" (out), "digit" (in), and "bad" (noise or not-a-symbol). The

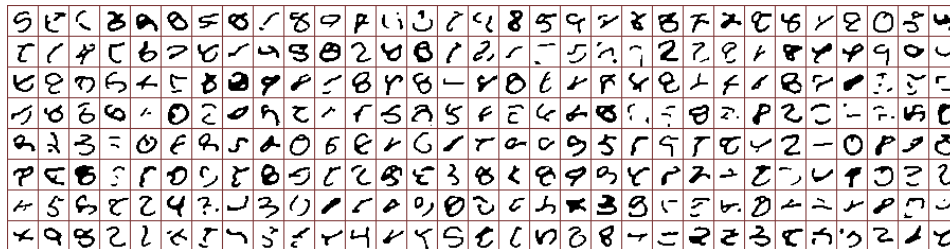

Figure 3: A random selection of symbols generated by one of our best sparse autoencoder, the same as the one that generated the letters in Figure 4(b).

| | inter-score correlations | | | | | | | | human counts | | |
| --- | --- | --- | --- | --- | --- | --- | --- | --- | --- | --- | --- |
| | oc | om | oo | op | ic | im | io | ip | out | in | bad |
| **out count** | 1 | -0.03 | -0.13 | 0.04 | -0.12 | 0.02 | -0.07 | -0.11 | 12 | 0 | 8 |
| **out max** | -0.03 | 1 | -0.07 | 0.01 | -0.16 | -0.10 | 0.03 | -0.09 | 15 | 0 | 5 |
| **out objectness** | -0.13 | -0.07 | 1 | 0.21 | -0.06 | 0.08 | 0.02 | -0.08 | 9 | 10 | 1 |
| **out Parzen** | 0.04 | 0.01 | 0.21 | 1 | -0.17 | 0.01 | -0.19 | -0.20 | 4 | 13 | 3 |
| **in count** | -0.12 | -0.16 | -0.06 | -0.17 | 1 | 0.30 | 0.1 | 0.14 | - | - | - |
| **in max** | 0.02 | -0.10 | 0.08 | 0.01 | 0.30 | 1 | 0.03 | 0.06 | - | - | - |
| **in objectness** | -0.07 | 0.03 | 0.02 | -0.19 | 0.1 | 0.03 | 1 | 0.00 | - | - | - |
| **in Parzen** | -0.11 | -0.09 | -0.08 | -0.20 | 0.14 | 0.06 | 0.00 | 1 | 0 | 17 | 3 |

Table 3: Inter-score correlations among top 10% models per score and human annotation counts among top twenty models per score. out=letters; in=digits.

last three columns of Table 3 show that the out-of-class count and out-of-class max scores work well in selecting good out-of-class generators, especially with respect to in-class generators. They are relatively bad in selecting good generators overall. Symmetrically, out-of-class objectness and the Parzen measures select, with high accuracy, good quality models, but they mix out-of-class and in-class generators (digits and letters). Parzen scores are especially bad at picking good out-of-class generators. Somewhat surprisingly, even out-of-class Parzen is picking digits, probably because in-distribution digit generators generate more regular, less noisy images than out-of-class letter generators. In other words, opening the space towards non-digit like "spurious" symbols come at a price of generating less clean symbols which are farther from letters (in a Parzen sense) than clean digits.

We also computed the inter-score correlations in the following way. We first selected the top 10% models for each score because we were after the correlation of the best-performing models . Then we computed the Spearman rank correlation of the scores (so we did not have to deal with different scales and distributions). The first eight columns of Table 3 show that i) in-class and out-of-class measures are anti-correlated, ii) out-of-class count and max are uncorrelated, and are somewhat anti-correlated with out-of-class objectness.

These results suggest that the best strategy is to use out-of-class objectness for selecting good quality models and out-of-class count and max to select models which generate letters. Figure 4 illustrates the results by pangrams (sentences containing all letters) written using the generated symbols. The models (a)-(d) were selected automatically: these were the four models that appeared in the top ten both according to out-of-class objectness and out-of-class counts. Letters of the last sentence (e) were hand-picked by us from letters generated by several top models. Among the four models, three were fully connected autoencoders with sparsity and one was a GAN. All of the three sparse autoencoders had five hidden layers and used a small noise corruption (less than 0.1). The GAN used the default learning rate of 0.0002 and a large number (2048) of fully connected hidden units for the generator, while the number of fully connected hidden units of the discriminator was significantly smaller (128).

(a) PACK MY BOX WITH FIVE DOZEN LIQUOR JUGS
(b) PACK MY BOX WITH FIVE DOZEN LIQUOR JUGS
(c) PACK MY BOX WITH FIVE DOZEN LIQUOR JUGS
(d) PACK MY BOX WITH FIVE DOZEN LIQUOR JUGS
(e) PACK MY BOX WITH FIVE DOZEN LIQUOR JUGS

Figure 4: Pangrams created (a-d) using top models selected automatically, and (e) using letters selected from several models by a human.

## 6    DISCUSSION AND PERSPECTIVES

In this paper we have proposed a framework for designing and analysing generative models for novelty generation. The quantitative measures make it possible to systematically study the creative capacity of generative models. We believe that human evaluation will remain an important source of feedback in this domain for the foreseeable future. Nevertheless, quantitative measures, such as our out-of-class objectness and out-of-class count and max, will i) make it possible to semi-automate the search for models that exhibit creativity, and ii) allow us to study, from the point of view of novelty generation, the numerous surrogates used for evaluating generative models (Theis et al., 2015), especially those that explicitly aim at quantifying creativity or interestingness (Schmidhuber, 2009).

The main focus of this paper was setting up the experimental pipeline and to analyze various quality *metrics*, designed to measure out-of-distribution novelty of samples and generative models. The immediate next goal is to analyze the *models* in a systematic way, to understand what makes them "memorizing" classes and what makes them opening up to generate valuable out-of-distribution samples.

## 7    ACKNOWLEDGMENTS

This work was partially supported by the HPC Center of Champagne-Ardenne ROMEO.

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
