# Peer review of "Out-of-class novelty generation: an experimental foundation"

_ICLR 2017 — rejected_

[Official Review · AnonReviewer1 · rating 5 · confidence 3 · 15 Dec 2016]
**No Title**

This paper proposed a quantitative metric for evaluating out-of-class novelty of samples from generative models. The authors evaluated the proposed metric on over 1000 models with different hyperparameters and performed human subject study on a subset of them.

The authors mentioned difficulties in human subject studies, but did not provide details of their own setting. An "in-house" annotation tool was used but it's unclear how many subjects were involved, who they are, and how many samples were presented to each subject. I'm worried about the diversity in the subjects because there may be too few subjects who are shown too many samples and/or are experts in this field.

This paper aims at proposing a general metric for novelty but the experiments only used one setting, namely generating Arabic digits and English letters. There is insufficient evidence to prove the generality of the proposed metric.

Moreover, defining English letters as "novel" compared to Arabic digits is questionable. What if the model generates Arabic or Indian letters? Can a human who has never seen Arabic handwriting tell it from random doodle? What makes English letters more "novel" than random doodle? In my opinion these questions are best answered through large scale human subject study on tasks that has clear real world meanings. For example, do you prefer painting A (generated) or B (painted by artist).

[Official Review · AnonReviewer2 · rating 4 · confidence 4 · 18 Dec 2016]
**No Title**

The authors proposed an way to measure the generation of out-of-distribution novelty. Their methods implied, if a model trained on MNIST digits could generate some samples are more like letters judged by anther model trained both on MNIST and letters,  the model trained on MNIST could be seen as having  the ability to generate novel samples. Some empirical experiments were reported. 
The novelty is hard to define. The proposed metric is also problematic. A naive combination of MNIST and letters dataset do not represent the natural distribution of handwritten digits and letters. IT means that the model trained on the combination could not properly distinguished digits and letters. The proposed out-of-class count and out-of-class max are thus pointless. For the "novel" samples in Fig. 3,  they are clearly digits. I guess they quantize the samples to binary. If they would quantize the samples to 8 bit, the resulting images would look even more like digits.

[Official Review · AnonReviewer3 · rating 6 · confidence 3 · 29 Dec 2016]
**Paper fights a difficult battle to defend and unify novelty generating models. It fights somewhat well.**

First, the bad:

This paper is frustratingly written. The grammar is fine, but:
 - The first four pages are completely theoretical and difficult to follow without any concrete examples. These sections would benefit greatly from a common example woven through the different aspects of the theoretical discussion.
 - The ordering of the exposition is also frustrating. I found myself constantly having to refer ahead to figures and back to details that were important but seemingly presented out of order. Perhaps a reordering of some details could fix this. Recommendation: give the most naturally ordered oral presentation of the work and then order the paper similarly.

Finally, the description of the experiments is cursory, and I found myself wondering whether the details omitted were important or not. Including experimental details in a supplementary section could help assuage these fears.

The good:

What the paper does well is to gather together past work on novelty generation and propose a unified framework in which to evaluate past and future models. This is done by repurposing existing generative model evaluation metrics for the task of evaluating novelty. The experiments are basic, but even the basic experiments go beyond previous work in this area (to this reviewer’s knowledge).

Overall I recommend the paper be accepted, but I strongly recommend rewriting some components to make it more digestible. As with other novelty papers, it would be read thoroughly by the interested few, but it is likely to fight an uphill battle against the majority of readers outside the sub-sub-field of novelty generation; for this reason the theory should be made even more intuitive and clear and the experiments and results even more accessible.

[Official Review · AnonReviewer4 · rating 7 · confidence 4 · 02 Jan 2017]

This paper examines computational creativity from a machine learning perspective. Creativity is defined as a model's ability to generate new types of objects unseen during training. The authors argue that likelihood training and evaluation are by construction ill-suited for out-of-class generation and propose a new evaluation framework which relies on the use of held-out classes of objects to measure a model's ability to generate new and interesting object types.

I am not very familiar with the literature on computational creativity research, so I can't judge on how well this work has been put into the context of existing work. From a machine learning perspective, I find the ideas presented in this paper new, interesting and thought-provoking.

As I understand, the hypothesis is that the ability of a model to generate new and interesting types we *do not* know about correlates with its ability to generate new and interesting types we *do* know about, and the latter is a good proxy for the former. The extent to which this is true depends on the bias introduced by model selection. Just like when measuring generalization performance, one should be careful not to reuse the same held-out classes for model selection and for evaluation.

Nevertheless, I appreciate the effort that has been made to formalize the notion of computational creativity within the machine learning framework. I view it as an important first step in that direction, and I think it deserves its place at ICLR, especially given that the paper is well-written and approachable for machine learning researchers.

[Final Decision · Program Chairs · 06 Feb 2017]
**ICLR committee final decision**

This paper aims to present an experimental framework for selecting machine learning models that can generate novel objects. As the work is devoted to a relatively subjective area of study, it is not surprising that opinions of the work are mixed.
 
 A large section of the paper is devoted to review, and more detail could be given to the experimental framework. It is not clear whether the framework can actually be useful outside the synthetic setup described. Moreover, I worry it encourages unhealthy directions for the field. Over 1000 models were trained and evaluated. There is no form of separate held-out comparison: the framework encourages people to keep trying random stuff until the chosen measure reports success.